# Influence of psychological capital on core competency for new nurses

**Junqiang Wang[1]ʘ, Jiya Chen[2]ʘ, Lingzhi Zheng[3], Baojin Zeng[1], Xiaoting Yan[4], Mengjie Xia**  **[5,6]\*, Lili Chen[2]\***

1 Department of Gynecology, Taizhou Hospital of Zhejiang Province Affiliated to Wenzhou Medical University, Taizhou, Zhejiang, China, 2 Department of Nursing, Taizhou Hospital of Zhejiang Province Affiliated to Wenzhou Medical University, Taizhou, Zhejiang, China, 3 Department of Obstetrics, Taizhou Hospital of Zhejiang Province Affiliated to Wenzhou Medical University, Taizhou, Zhejiang, China, 4 Central Sterile Supply Department, Taizhou Central Hospital (Taizhou University Hospital), Taizhou, Zhejiang, China, 5 School of Medicine, Taizhou University, Taizhou, Zhejiang, China, 6 School of Nursing, Faculty of Medicine Bioscience & Nursing, MAHSA University, Jenjarom, Selangor, Malaysia

ʘ These authors contributed equally to this work.
\* xmj1140590151@outlook.com (MX); chenlili@enzemed.com (LC)

**Data Availability Statement:** All relevant data are within the paper and its Supporting Information files.

**Funding:** -C. L. -NO.22EZB14 -Taizhou Enze Medical Center (Group) Scientific Research Project

## Abstract

### Background

The development of core competency is crucial for the success of new nurses, enabling them to deliver high-quality care. Psychological capital (PsyCap), encompassing self-efficacy, optimism, hope, and resilience, significantly influences individuals' abilities and achievements across various professions. However, limited research has specifically examined the impact of PsyCap on the core competency of new nurses. This study aims to bridge this gap by investigating the relationship between PsyCap and core competency development in new nurses, providing valuable strategic insights for improving PsyCap and promoting core competence acquisition.

### Methods

142 new nurses were chosen for the investigation using a convenient cluster sampling method. The questionnaire included components on socio-demographic characteristics, the Competency Inventory for Registered Nurses (CIRN), and the PsyCap Questionnaire-24 (PCQ-24). The $t$-test, One-Way ANOVA, Pearson correlation analysis and hierarchical multiple regression were used for statistical analysis.

### Result

The number of valid questionnaires was 138, and the effective return rate was 97.2%. The overall mean score for core competencies was 171.01 (SD 25.34), and the PsyCap score was 104.76(SD 13.71). The PsyCap of new nurses was highly correlated with core competency, with a correlation coefficient of r = 0.7, p < 0.01. Self-efficacy of PsyCap is a significant independent predictor of core competency (adjust $R^2$ = 0.49).

Fund -This is a hospital-funded project, so there is no URL -The funders play any role in the data collection and decision to publish.

**Competing interests:** The authors have declared that no competing interests exist.

## Conclusion

Self-efficacy in PsyCap is an important predictor of new nurses' core competency. Nursing managers should pay sufficient attention to the cultivation and development of new nurses' Psy-Cap, with particular emphasis on enhancing self-efficacy to improve their core competency.

## Introduction

The concept of competency was first introduced by Harvard psychologist McClelland in 1973 [1], and it is defined by the Australian Institute of Nurses and Midwives as the combination of skills, knowledge, attitudes, values, and abilities demonstrated in the professional field [2]. The International Council of Nurses (ICN) proposed a core competency framework for registered nurses in 2003, emphasizing the knowledge, skills, judgment, and personal attributes required to provide safe and ethical nursing care [3]. The American Association of Colleges of Nursing (AACN) identified 4 core nursing competencies as critical thinking, assessment, communication, and technical operations [4]. International organizations such as the World Health Organization (WHO) emphasize the importance of qualified nurses to the safety and quality of the healthcare system, and that low nurse core competency increases inpatient morbidity and mortality [5]. As new nurses play a vital role in the nursing team, their core competencies are particularly important [6].

Psychological capital (PsyCap) refers to an individual's positive mental state during growth and development, encompassing self-efficacy, optimism, resilience, and hope [7] It is an important resource for individuals to effectively cope with work stress and reduce burnout [8,9]. Previous research has linked PsyCap to job performance and job satisfaction among healthcare professionals [10] and clinical decision-making skills among nurses [11]. Understanding the relationship between PsyCap and core competencies in nursing practice is crucial.

New nurses face various challenges during the transition from education to the clinical setting, including adapting to new protocols, managing increased responsibilities, and coping with emotional pressures [12]. Understanding the factors that contribute to their core competency and PsyCap is essential for their successful transition and retention in the nursing profession. The local setting significantly influences the development of core competencies and PsyCap among new nurses, with organizational culture, leadership support, workload, and interpersonal relationships playing important roles [13,14]. Exploring these factors within the local context is vital for identifying specific challenges and opportunities and developing targeted interventions and support systems.

While many studies have investigated various variables influencing core competency development among new nurses, the role of PsyCap remains understudied. The extent to which PsyCap affects new nurses' core competency is currently unclear [15–17]. This study aims to fill this gap by examining the relationship between core competency and PsyCap among new nurses within the local healthcare system. The findings will provide valuable insights for nurse educators, healthcare administrators, and policymakers in creating an environment that supports the growth and well-being of new nurses by addressing the challenges and opportunities that impact their competency development and PsyCap.

### Research framework

The research framework employed in this study is based on the well-established Job Demands-Resources (JD-R) model, which offers valuable insights into the relationship between new

nurse core competencies and PsyCap [18]. The JD-R model posits that job demands and resources significantly impact employee well-being and performance. Within this framework, job demands encompass workload, time pressure, emotional demands, and organizational demands [19]. On the other hand, job resources encompass a supportive work environment, opportunities for professional development, and feedback and recognition [20]. PsyCap, consisting of hope, optimism, self-esteem, and resilience, is an important psychological resource [21]. The core competencies of new nurses, including clinical knowledge, communication skills, critical thinking abilities, and teamwork, are the key dependent variables of interest [22]. By examining the associations between job demands, job resources, PsyCap, and core competencies, this study aims to enhance our understanding of the factors influencing new nurse performance and well-being.

## Hypotheses

Based on the research framework, the following research hypotheses are proposed:

$H_1$: There is a positive relationship between PsyCap and core competencies among new nurses.

$H_2$: Socio-demographic characteristics (such as gender, age, and educational level) are related to differences in core competencies among new nurses.

$H_3$: PsyCap significantly predicts core competencies among new nurses, even after controlling for socio-demographic characteristics.

$H_4$: The four dominions of PsyCap, namely hope, optimism, self-efficacy, and resilience, significantly predict the level of core competencies among new nurses.

These hypotheses guide the investigation and provide specific expectations for the relationships between PsyCap, core competencies, and socio-demographic characteristics among new nurses. The study aims to test these hypotheses and contribute to the understanding of the role of PsyCap in the development of core competencies in the nursing profession.

## Methods

### Study design

A cross-sectional study design was employed to investigate the relationship between core competency and psychological capital among new nurses.

### Study population

In this study, new nurses within three years of training were defined as "new nurses" to reflect their status during the training cycle. A convenient cluster sampling method was used to select potential study participants from a tertiary hospital in a coastal area of China. The inclusion criteria for new nurses included: (1) holding a nursing license, (2) have no prior work experience as a registered nurse or any other healthcare-related role, and (3) voluntary participation.

### Data collection

Data collection was conducted between January and February 2023. The researchers initially contacted the nursing department of the hospital to obtain approval for the study. An online questionnaire, along with an explanatory statement, was sent to the nurses using the Questionnaire Star software. The questionnaire was filled anonymously and independently, allowing

for only one response per IP address. The collected data were exported and stored in an Excel database, ensuring confidentiality with password-protected access.

## Measurements

The online self-rating questionnaire consisted of socio-demographic characteristics, the Competency Inventory for Registered Nurses (CIRN), and the PsyCap Questionnaire-24 (PCQ-24). Socio-demographic variables included gender, length of professional service, marital status, childbirth status, educational background, and nature of employment.

New nurses' core competency was measured by the CIRN. The scale developed by Liu Ming et al [22], has been widely applied to investigate the level of the core competency of nurses [23,24]. It has 58 items over seven categories: critical thinking and research aptitude(10 items), clinical care(9 items), leadership(10 items), interpersonal relationships(8 items), legal/ethical practice(8 items), professional development(6 items), and teaching-coaching(7 items), measured on a 5-point Likert scale (0 = not at all competent, 4 = very competent), higher scores indicate that the nurse has greater core competencies. In our study, the same scale demonstrated excellent internal consistency, with a Cronbach's α coefficient of 0.98, indicating strong agreement among the 58 items. These findings support the originality and reliability of the scale in assessing the core competency of new nurses. The scale's validity was confirmed by a Kaiser-Meyer-Olkin (KMO) value of 0.88, indicating high intercorrelation among the scale items. Furthermore, Bartlett's test of sphericity yielded a significant chi-square value of 10218.35 (df = 1653, p < 0.001), indicating substantial interrelationships among the scale items and supporting the construct validity of the scale.

The PCQ-24 scale was used to measure the responders' positive psychological capital (PsyCap). It consists of four dimensions: self-efficacy, hope, resilience, and optimism, with each dimension comprising six items [21] The items were assessed using a 6-point Likert scale, ranging from 1 (strongly disagree) to 6 (strongly agree). Cumulative scores for the PCQ and its individual measures were calculated by summing the included items, with higher scores indicating a greater level of PsyCap. The scale demonstrated high internal consistency, with a Cronbach's α coefficient of 0.93. Additionally, it showed excellent sampling adequacy, with a Kaiser-Meyer-Olkin (KMO) value of 0.94, and the Bartlett's test of sphericity was highly significant ($\chi^2$ = 3713.26, df = 276, p < 0.001), indicating strong evidence of construct validity and relationships among the variables assessed.

## Ethical consideration

The study was carried out with the agreement of the hospital's Human Research Ethics Committee(No. K20230103). Before participants get the online questionnaire, they were informed of the study's objective and procedure then deciding whether or not to complete it. Participants were informed of their right to voluntarily decide whether to participate in the study without facing any adverse consequences. Once the participants consented to participate, the researchers proceeded to send them the questionnaire. The information was gathered anonymously and kept private. Neither during nor after data collection could any writers access information that may individually identify individuals. The data were kept on a computer that was solely accessible to the study team.

## Data analyses

SPSS 26.0 was employed. A two-sided with p<0.05 was applied to all data analysis. The Kolmogorov-Smirnov test was used to analyze the normal distributions for both PsyCap and core competencies. The difference in core competencies across socio-demographic characteristics

was investigated using t-test and One-Way ANOVA. PsyCap and core competencies were described using means and standard deviations. The relationship between core competencies and PsyCap was evaluated using hierarchical multiple regression and Pearson correlation analysis.

## Results

Table 1 presents the socio-demographic characteristics of the study participants (n = 138). The majority of participants were female (93.48%), single (93.48%), and without children (97.10%). Regarding professional experience, 34.06% had one year of service, 24.63% had two years, and 41.30% had three years. Approximately 46.38% held a diploma, while 44.20% had a bachelor's degree. The participants mostly worked as contract nurses (75.36%). Statistical analyses, including one-way ANOVA and t-tests, revealed no significant differences in core competencies based on gender, professional experience, marital status, childbirth status, education, or employment nature ($p > 0.05$).

Table 2 summarizes the mean scores and standard deviations of the core competencies and psychological capital variables. The overall mean score for core competency was 171.01 (SD 25.34), indicating a medium-high level among new nurses. Among the core competency dimensions, teaching-coaching had the lowest mean score (M 2.8, SD 0.58), while legal/ethical practice had the highest (M 3.19, SD 0.62). New nurses exhibited a medium-high level of

**Table 1. Socio-demographic characteristics of new nurses.**

| Socio-demographic characteristics | N (%) | CIRN M (SD) | p |
|---|---|---|---|
| Gender | | | |
| Male | 9 (6.52%) | 169.78 (13.59) | 0.88 |
| Female | 129 (93.48%) | 171.09 (25.99) | |
| Length of professional service (years) | | | |
| One year | 47 (34.06%) | 172.96 (26.03) | 0.80 |
| Two years | 34 (24.63%) | 169.53 (24.50) | |
| Three years | 57 (41.30%) | 170.28 (25.59) | |
| Marital status | | | |
| Single | 129 (93.48%) | 170.49 (26.00) | 0.66 |
| Married | 8 (5.80%) | 178.88 (10.92) | |
| Divorced | 1 (0.72%) | 175.00 (0.00) | |
| Childbirth status | | | |
| No child | 134 (97.10%) | 170.91 (25.70) | 0.95 |
| One child | 3 (2.17%) | 172.67 (3.22) | |
| Two children and above | 1 (0.72%) | 179.00 (0.00) | |
| Educational background | | | |
| Diploma | 64 (46.38%) | 169.78 (26.77) | 0.79 |
| Associate degree | 13 (9.42%) | 174.85 (17.10) | |
| Bachelor degree | 61 (44.20%) | 171.48 (25.49) | |
| Nature of employment[a] | | | |
| Contract nurses | 104 (75.36%) | 171.00 (27.75) | 1.00 |
| Permanent nurses | 34 (24.64%) | 171.03 (16.16) | |

CIRN, Competency Inventory for Registered Nurse; M, Means; SD, Standard Deviations.

[a] Permanent nurses are permanent employees of healthcare facilities, enjoying benefits, job security, and career opportunities. Contract nurses, however, work temporarily and may have less job security and benefits. In some cases, contract nurses may perform similar duties as staff nurses but with different employment arrangements.

**Table 2. Descriptive analyses of core competencies and psychological capital (n = 138).**

| Variables | Min. | Max. | Total Score (M) | Total Score (SD) | Mean Score (M) | Mean Score (SD) |
|---|---|---|---|---|---|---|
| Competency Inventory for Registered Nurses | | | | | | |
| Critical thinking and research aptitude | 2 | 40 | 28.03 | 5.65 | 2.81 | 0.71 |
| Clinical care | 5 | 36 | 26.28 | 4.01 | 2.93 | 0.6 |
| Leadership | 10 | 40 | 29.34 | 5.56 | 2.93 | 0.72 |
| Interpersonal relationships | 8 | 32 | 24.55 | 3.65 | 3.05 | 0.53 |
| Legal/ethical practice | 8 | 32 | 25.49 | 4.16 | 3.19 | 0.62 |
| Professional development | 6 | 24 | 17.83 | 3.06 | 3.03 | 0.58 |
| Teaching–coaching | 1 | 28 | 19.49 | 4.74 | 2.8 | 0.58 |
| Total score | 46 | 232 | 171.01 | 25.34 | 2.96 | 0.68 |
| Psychological capital | | | | | | |
| Self-efficacy | 12 | 36 | 26.44 | 4.37 | 4.4 | 0.82 |
| Hope | 12 | 36 | 27.53 | 4.20 | 4.53 | 0.8 |
| Resilience | 11 | 34 | 25.86 | 3.42 | 4.42 | 1.04 |
| Optimism | 10 | 35 | 24.94 | 3.74 | 4.08 | 1.23 |
| Total score | 46 | 133 | 104.76 | 13.71 | 4.36 | 1 |

psychological capital (M 104.76, SD 13.71). The highest dimension of psychological capital was hope (M 27.53, SD 4.20), while optimism had the lowest score (M 24.94, SD 3.74).

Table 3 presents the correlations between psychological capital and core competencies. Significant positive correlations ($p < 0.01$) were found between all core competencies and psychological capital dimensions. Critical thinking and research aptitude demonstrated moderate positive correlations with self-efficacy, hope, resilience, and a weaker correlation with optimism. Clinical care showed strong positive correlations with all dimensions of psychological capital. Similarly, leadership, interpersonal relationships, legal/ethical practice, professional development, and teaching-coaching exhibited moderate to strong positive correlations with psychological capital. The total scores of core competencies were significantly positively correlated with the total scores of psychological capital, indicating a strong overall relationship between the two constructs.

A hierarchical multiple regression analysis was conducted to investigate the relationship between core competencies and psychological capital (Table 4). The model explained approximately 49% of the variance in the core competency total score. Among the predictor variables, efficacy total score showed a significant positive relationship with the core competency total score ($β = 0.27$, $p < 0.05$), indicating that higher levels of efficacy were associated with higher

**Table 3. Correlation analyses (R-values) between core competencies and psychological capital (n = 138).**

| | Self-efficacy | Hope | Resilience | Optimism | Total score |
|---|---|---|---|---|---|
| Critical thinking and research aptitude | 0.48[b] | 0.48[b] | 0.49[b] | 0.33[b] | 0.51[b] |
| Clinical care | 0.60[b] | 0.61[b] | 0.58[b] | 0.45[b] | 0.64[b] |
| Leadership | 0.56[b] | 0.50[b] | 0.45[b] | 0.30[b] | 0.52[b] |
| Interpersonal relationships | 0.55[b] | 0.57[b] | 0.59[b] | 0.42[b] | 0.61[b] |
| Legal/ethical practice | 0.46[b] | 0.51[b] | 0.53[b] | 0.42[b] | 0.55[b] |
| Professional development | 0.66[b] | 0.63[b] | 0.57[b] | 0.35[b] | 0.64[b] |
| Teaching–coaching | 0.63[b] | 0.56[b] | 0.50[b] | 0.33[b] | 0.59[b] |
| Total score | 0.68[b] | 0.66[b] | 0.63[b] | 0.44[b] | 0.70[b] |

[b] P<0.01.

**Table 4. Hierarchical multiple regression for core competency and psychological capital (n = 138).**

| Variables | B | β | t | p | $R^2$ | Adjusted $R^2$ | F |
|---|---|---|---|---|---|---|---|
| Constant term | 39.9 | - | 3.23 | 0.00[c] | 0.51 | 0.49 | F = 34.06 p = 0.000 [c] |
| Efficacy total score | 1.54 | 0.27 | 2.26 | 0.03[d] | | | |
| Hope total score | -0.37 | -0.06 | -0.42 | 0.68 | | | |
| Resilience total score | 0.06 | 0.01 | 0.07 | 0.95 | | | |
| Optimism total score | -0.23 | -0.03 | -0.42 | 0.68 | | | |
| PCQ total score | 1 | 0.54 | 10.33 | 0.00 [c] | | | |

[c] $p < 0.01$, [d] $p < 0.05$.

levels of core competencies. However, hope, resilience, and optimism total scores did not show statistically significant associations with the core competency total score. The psychological capital total score demonstrated a significant positive relationship with the core competency total score (β = 0.54, $p < 0.01$), suggesting that higher levels of psychological capital were associated with higher levels of core competencies among new nurses.

## Discussion

The present study fills a significant gap in the existing literature by exploring the relationship between psychological capital (PsyCap) and the core competency of new nurses. The findings emphasize the originality and uniqueness of this study. The evaluation of core competency among new nurses in this study indicates a good level of competence, which is consistent with similar studies in Norway [25] and slightly higher than that of Chinese another study [24]. This finding highlights the potential universality of core competency among new nurses, irrespective of cultural and geographical variations. Secondly, this investigation found that the mean value of the new nurses' PsyCap scores for each item was 4.36 (SD 1), which was higher than the theoretical median (3.5). The overall PsyCap of new nurses is at a moderate to high level, consistent with the findings of Jiao Yang et al [26]. This consistency suggests that PsyCap may play a vital role in the nursing profession, regardless of the specific sample or study design. However, it is worth noting that this study expands upon previous research by specifically investigating the relationship between PsyCap and core competency. By focusing on the interplay between these two constructs, this study sheds light on the potential influence of psychological resources on the development of nursing competence, offering a fresh perspective within the field.

In terms of core competency, the participants rated their legal/ethical practice ability highest, while their teaching-coaching capacity received the lowest ratings. The participants demonstrated a strong sense of competence in respecting patient privacy, upholding patient rights, and fulfilling their nursing obligations [27,28]. However, they perceived themselves to be less competent in identifying the root cause of patients' illnesses, making rational judgments based on patient conditions, and analyzing patients' needs from different perspectives [29,30]. Furthermore, the participants identified difficulties in assisting colleagues to adapt quickly to their new roles and faced challenges in their ability to educate patients and families about their condition [22].

Regarding PsyCap, hope received the highest score, while optimism received the lowest score. This indicates that although new nurses are positively motivated by their determination to accomplish their goals and their methods and abilities to achieve them, they lack positive attributions about their present and future success [7,31].

The significant and positive connections discovered between PsyCap and core competencies, including self-efficacy, hope, resilience, and optimism, are in line with the findings of Luthans et al. [32] These results highlight the potential impact of PsyCap on performance improvement. PsyCap serves as a positive psychological state that stimulates positive behaviors among individuals to fulfill their job tasks [32]. A higher level of PsyCap can enable new nurses to maintain a positive and optimistic work attitude, think critically in their work, and facilitate the development of clinical thinking skills, thus promoting their professional growth [33]. Nurses with higher levels of PsyCap are more adept at coping with work-related stress, finding solutions in challenging situations, and are more likely to experience higher job satisfaction and adaptability, resulting in enhanced competence [34]. Therefore, nursing managers should focus on the cultivation and utilization of nurses' motivation [35]. Strategies such as case sharing, guidance, and encouragement can be employed to enhance nurses' PsyCap [3,36]. New nurses should also continue their professional development, increase their knowledge base, and improve their PsyCap by fostering resilience, tenacity, and maintaining an optimistic and positive mindset [37,38].

Regression analysis results indicate that self-efficacy is a significant predictor variable for the core competencies of new nurses. This underscores the essential role of self-efficacy as a core competency quality for new nurses. Studies consistently demonstrate that enhancing nurses' self-efficacy can improve their performance and overall core competencies [39]. Therefore, nursing managers should prioritize the cultivation of new nurses' self-efficacy by guiding them to face stress and improving their self-efficacy from the perspective of positive psychology, thereby promoting career development and enhancing overall core competencies [40,41].

While this study contributes valuable insights, certain limitations should be considered. Firstly, the cross-sectional design restricts the interpretation of causal links between variables. Additionally, the results may not be entirely representative of mainland China, as nurses were solely recruited from hospitals. Furthermore, the use of self-report measures may have influenced the impartiality and reliability of the gathered information. Moreover, the examination of only a few sociodemographic characteristics as potential predictors of core competencies calls for further investigation of additional factors to gain a comprehensive understanding of core competencies. It is recommended that future studies employ longitudinal research designs, mixed-method approaches, and random sampling to better represent the target population and explore a wider range of individual and organizational-related factors.

## Conclusion

This study highlights the importance of the core competency and PsyCap of new nurses. The findings emphasize the positive impact of PsyCap on the core competencies of new nurses, regardless of cultural and geographical variations. Nursing managers and relevant institutions should recognize the significance of investing in the development of PsyCap and core competencies in new nurses to promote their professional growth and overall competence. Future research should consider exploring the variations in the core competencies of new nurses within different local contexts and investigate additional factors that may influence their core competencies. It is important to acknowledge the limitations of this study, such as the specific sample and research scope, and further research should address these limitations to enhance the understanding of new nurses' core competencies.

## Supporting information

**S1 Checklist. STROBE statement—checklist of items that should be included in reports of observational studies.**
(DOCX)

**S1 Data.**
(XLSX)

## Acknowledgments

We would like to thank the Nursing Department at Enze Hospital for their assistance in providing the online questionnaire. Also, we express our gratitude to all nurse participants.

## Author Contributions

**Conceptualization:** Mengjie Xia.

**Formal analysis:** Lingzhi Zheng, Baojin Zeng.

**Funding acquisition:** Lili Chen.

**Investigation:** Junqiang Wang, Jiya Chen, Lili Chen.

**Methodology:** Junqiang Wang, Jiya Chen.

**Project administration:** Lili Chen.

**Resources:** Jiya Chen, Lingzhi Zheng.

**Software:** Baojin Zeng.

**Supervision:** Mengjie Xia, Lili Chen.

**Validation:** Lingzhi Zheng.

**Writing – original draft:** Junqiang Wang, Jiya Chen.

**Writing – review & editing:** Junqiang Wang, Xiaoting Yan, Mengjie Xia.

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
