## [Decision Letter · Decision Letter 0]

6 Jun 2023

PONE-D-23-09529The Influence of psychological capital on core competency for new nursesPLOS ONE

Dear Dr. Xia,

Thank you for submitting your manuscript to PLOS ONE. After careful consideration, we feel that it has merit but does not fully meet PLOS ONE’s publication criteria as it currently stands. Therefore, we invite you to submit a revised version of the manuscript that addresses the points raised during the review process.

We look forward to receiving your revised manuscript.

Kind regards,

Chunyu Zhang

Academic Editor

PLOS ONE

3. Please include your tables as part of your main manuscript and remove the individual files. Please note that supplementary tables (should remain/ be uploaded) as separate "supporting information" files

Reviewers' comments:

Reviewer's Responses to Questions

**Comments to the Author**

1. Is the manuscript technically sound, and do the data support the conclusions?

Reviewer #1: Partly

Reviewer #2: Yes

Reviewer #3: Partly

2. Has the statistical analysis been performed appropriately and rigorously? 

Reviewer #1: No

Reviewer #2: Yes

Reviewer #3: No

3. Have the authors made all data underlying the findings in their manuscript fully available?

Reviewer #1: No

Reviewer #2: Yes

Reviewer #3: No

4. Is the manuscript presented in an intelligible fashion and written in standard English?

Reviewer #1: Yes

Reviewer #2: Yes

Reviewer #3: Yes

5. Review Comments to the Author

Reviewer #1: I reviewed a piece of work entitled the influence of psychological capital on core competency for new nurses. The competency of new nurses is a fascinating topic to address, but the study needs a solid, fundamental justification. To improve the quality of the manuscript's content overall, a few aspects need to be thoroughly examined.

The manuscript needs to be strictly followed and the instructions returned by the journal.

The abstract's content, particularly the study's aim and background, needs to be reorganized. It is good to combine the two. Please describe the sorts of ANOVA analysis that were employed in the method.

Additional information is needed in the introduction to strengthen the situation for why new nurses should be studied on the relationship between their core competency and psychological capital. Are there any important problems that should be brought up in relation to the competency and psychological capital of new nurses in the local setting?

The methodology requires extra info to make robust. The study was conducted in a single institution which might not reflect the entire target population in China. The selection of the sample applies a convenient sampling method which could further increase the selection bias. No sample size calculation was explained. Both details of the questionnaires are required especially on their originality of the validity and reliability of the tool. The scoring system applied in each of them, and the detail of the domain and items included.

The explanation on the application of the hierarchy multiple regression analysis probably needed. It does not reflect the true picture of analysis if we refer the Table 4.

The results of the study should be self-explanatory. Table 1 – how did you define the “new nurses“ when you include those who already three years in service? Nature of the nurses – how did you define it? What is the difference between contract nurses and staff nurses?

Table 2 – it is good if you include the minimum and maximum value of the score for each item. What is the purposes of including Total score M and Total score SD?

Table 3 – the p value at the bottom of the table refers to which correlation analysis (which pairing)? Are you saying that all pairs are not statistically significant? It does not correspond well with the description.

Table 4 – the hierarchy multiple regression analysis needs to be shown clearly step by step.

The overall discussion probably need to be rewrite following the correction in the method and result section.

Reviewer #2: 1. This paper presents an empirical and rigorous investigation, yielding analysis results of substantial research reference value. However, it is imperative to enhance and refine the motivation for conducting such research. In the "Background" section, the author acknowledges the current lack of clarity regarding the extent to which psychological capital influences the core competence of new nurses. Nonetheless, the final "Discussion" reveals consistency with previous research.

For instance:

1. This result is similar to that for newly graduated nurses in Norway (15) but slightly higher than that of Chinese another study (13).

2. The overall psychological capital of new nurses is at a moderate to the high level, which is consistent with the findings of Jiao Yang et al (16).

In light of these findings, it becomes imperative to underscore the significance and distinctiveness of this paper.

2. In order to enhance the readability of the article for an international journal, it may be beneficial to incorporate additional content. Consider augmenting the existing information or introducing research questions/research hypotheses in the literature section.

3. Furthermore, it is worth considering the inclusion of a research framework within your study. This framework can provide a structured and comprehensive approach to guide the investigation and strengthen the overall research methodology.

Reviewer #3: Dear Author,

Your article is interesting. Naturally, there are parts that can be revised. However, there is an important problem that is not very possible to revise, and that creates the wrong opinion about the selection of new nurses. Below is my assessment of each part of your article.

The title effectively conveys the main topic of the research in a clear and concise manner. The summary is clear, understandable and clear.

The introduction explains core competency and psychological capital for nurses, but does not explain the rationale for researching the effect of psychological capital on core competeny. The results of other studies of new nurses that include variables that may affect core competency should be included and should convince the reader that the effect of psychological capital should be investigated.

In the method, the second criterion specified as the selection criterion of the participants is not clear. It would be appropriate to choose a time limit reported in the literature. The most important element to be defined in this study. In the literature, new nurses are defined as nurses who have joined an organization within the last 30 days. I would like to know the mean and standard deviations of the study times of the participants. It is not clear whether there are nurses who have worked for a few months in the group whose working period is specified as one year, or whether there are nurses who have completed a full year. For the nurse who has worked for two or three years, proof of the "new nurses" designation should have been added. The population and sample size of the study is insufficient. Power analysis is also recommended. It is not clear whether the Cronbach's α values given for the scales used belong to the current study.

Ethics committee approval is indicated, but information on how and how informed consent was obtained should be included.

Hierarchical preference of regression analysis is not appropriate in the analysis of data. While a significant relationship was found between all subscales of psychological capital and core competency, it is not possible to explain the reason for including only self-efficacy in the analysis.

In the discussion section, it is seen that comments and inferences are missing, except for mentioning the similarity with previous studies. In line with the purpose of the study, there is a need to deepen the discussion.

6. PLOS authors have the option to publish the peer review history of their article (what does this mean?). If published, this will include your full peer review and any attached files.

Reviewer #1: No

Reviewer #2: No

Reviewer #3: No

---

## [Author Response · Author response to Decision Letter 0]

4 Jul 2023

Dear Editor and Reviewers:

Thank you for your comments concerning our manuscript entitled “Influence of psychological capital on core competency for new nurses”. Those comments are all valuable and very helpful for revising and improving our paper, as well as the important guiding significance to our research. We have studied the comments carefully and have made corrections which we hope meet with approval. Revised portions are highlighted on the paper. The main corrections in the paper and the responses to the reviewer’s comments are as flowing: 

Response to the reviewer’s comments: 

Reviewer #1:

1. Response to comment: The manuscript needs to be strictly followed and the instructions returned by the journal.

Response: We have adjusted the manuscript format to comply with the journal's requirements as per your suggestion.

2. Response to comment: The abstract's content needs to be reorganized. Please describe the sorts of ANOVA analysis that were employed in the method.

Response: We have reorganized the abstract's content and provided a description of the one-way ANOVA analysis used in the methodology. 

3. Response to comment: Additional information is needed in the introduction.

Response: Additional information has been added to the introduction section, focusing on the development of core competency and psychological capital in new nurses. We have highlighted the changes made, emphasizing the importance of understanding factors such as challenges during the transition, organizational culture, leadership support, workload, and interpersonal relationships..

4. Response to comment: The methodology requires extra info to make robust.

Response: We included all new nurses entering in 2020-2022 and did not calculate a sample size. We further stated in the study limitations that the use of convenience sampling was motivated by accessibility and usability considerations and that alternative sampling methods, such as random sampling, could be discussed and considered for better representation of the target population. It is recommended that the results be considered as preliminary findings and future studies are encouraged to validate them on a larger sample and more representative sample. In addition, we have added a description of the validity and reliability of the originality of the questionnaire, as well as a description of the scoring system and the detailed domains.

5. Response to comment: The explanation on the application of the hierarchy multiple regression analysis

Response: We have provided a more specific explanation of the hierarchical multiple regression analysis, highlighting its relevance and implications.

6. Response to comment: Table 1 – how did you define the “new nurses” when you include those who already three years in service? Nature of the nurses – how did you define it? What is the difference between contract nurses and staff nurses?

Response: The definition of a new nurse may vary by country, region and healthcare institution. In China, nurses are often required to undergo three years of nursing training, including theoretical studies and practical training. Therefore, defining nurses within three years of training as new nurses may reflect their status during the training cycle. Staff nurses are permanent employees of healthcare facilities, enjoying benefits, job security, and career opportunities. Contract nurses, however, work temporarily and may have less job security and benefits. In some cases, contract nurses may perform similar duties as staff nurses but with different employment arrangements. To reduce ambiguity, we changed the staff nurse to a permanent nurse

7. Response to comment: Modifications to table 2

Response: We added reporting on the minimum and maximum values. The total dimensional score reflects the overall score of the participants on each dimension, and it helps us to understand the performance of the participants on the whole scale. And the dimensional mean scores allow us to understand the average level of participants' performance on each dimension by calculating the mean of the scores on each dimension.

8. Response to comment: Modifications to table 3

Response: e have indicated that "p" represents the 1% level of significance in Table 3.

9. Response to comment: Modifications to table 4

Response: We have made revisions to Table 4, redescribed the section on hierarchical multiple regression analysis, and modified the corresponding discussion.

Reviewer #2:

1. Response to comment: Underscore the significance and distinctiveness of this paper.

Response: We have emphasized the significance and distinctiveness of our paper in the discussion section, highlighting the focus on examining the relationship between psychological capital and core competency and the fresh perspective it offers within the field.

2. Response to comment: Augmenting the existing information or introducing research questions/research hypotheses in the literature section

Response: Four related hypotheses have been added to the literature section, augmenting the existing information.

3. Response to comment: Inclusion of a research framework within your study

Response: The research framework has been included after the introduction, as per your suggestion.

Reviewer #3:

1. Response to comment: Explain the rationale for researching the effect of psychological capital on core competency and the effect of psychological capital should be investigated.

Response: We have further explained the rationale for researching the effect of psychological capital on core competency and its importance in nursing professionals' development. The purpose and significance of the study have been elaborated upon in the introduction section.

2. Response to comment: The second criterion specified as the selection criterion of the participants is not clear

Response: The selection criterion for participants has been clarified, stating that "new nurses" are those who have no prior work experience as a registered nurse or in any other healthcare-related role.

3. Response to comment: The mean and standard deviations of the study times of the participants. Proof of the "new nurses" designation should have been added. 

Response: The definition of a new nurse may vary by country, region and healthcare institution. In China, nurses are often required to undergo three years of nursing training, including theoretical studies and practical training. Therefore, defining nurses within three years of training as new nurses may reflect their status during the training cycle. In China, the new graduate nurses every entry period is July-August. The shortest duration of work for the population in this study was 6 months (34.06%) and the longest was 2.5 years (41.30%). The mean of the working time of the population was 1.23 (SD 0.80).

4. Response to comment: Power analysis is also recommended. It is not clear whether the Cronbach's α values given for the scales used belong to the current study.

Response: While a power analysis was not necessary due to considering the entire population, we have acknowledged the limitations of convenience sampling and suggested alternative sampling methods for future studies. The validity and reliability of the questionnaire have been described, and the scoring system and detailed domains have been provided.

5. Response to comment: Ethics committee approval is indicated, but information on how and how informed consent was obtained should be included.

Response: We have added information on how informed consent was obtained, ensuring participants' voluntary participation without adverse consequences.

6. Response to comment: Hierarchical preference of regression analysis is not appropriate in the analysis of data.

Response: The section on hierarchical multiple regression analysis has been further refined, addressing your concern. Table 4 has been revised accordingly.

7. Response to comment: In line with the purpose of the study, there is a need to deepen the discussion.

Response: We have deepened the discussion section, highlighting the focus of our study on examining the relationship between psychological capital and core competency and the potential influence of psychological resources on nursing competence.

We believe that these revisions have significantly enhanced the clarity, accuracy, and overall quality of our manuscript. We are grateful for your guidance, which has undoubtedly strengthened our research. If you have any further suggestions or require additional information, please do not hesitate to let us know.

Thank you once again for your time and invaluable input. We look forward to the opportunity to submit the revised manuscript.

Best regards,

Mengjie Xia

---

## [Decision Letter · Decision Letter 1]

12 Jul 2023

Influence of psychological capital on core competency for new nurses

PONE-D-23-09529R1

Dear Dr. Xia,

We’re pleased to inform you that your manuscript has been judged scientifically suitable for publication and will be formally accepted for publication once it meets all outstanding technical requirements.

Kind regards,

Chunyu Zhang

Academic Editor

PLOS ONE

Additional Editor Comments (optional):

Reviewers' comments:

Reviewer's Responses to Questions

**Comments to the Author**

1. If the authors have adequately addressed your comments raised in a previous round of review and you feel that this manuscript is now acceptable for publication, you may indicate that here to bypass the “Comments to the Author” section, enter your conflict of interest statement in the “Confidential to Editor” section, and submit your "Accept" recommendation.

Reviewer #1: All comments have been addressed

Reviewer #2: All comments have been addressed

2. Is the manuscript technically sound, and do the data support the conclusions?

Reviewer #1: Yes

Reviewer #2: Yes

3. Has the statistical analysis been performed appropriately and rigorously? 

Reviewer #1: Yes

Reviewer #2: Yes

4. Have the authors made all data underlying the findings in their manuscript fully available?

Reviewer #1: Yes

Reviewer #2: Yes

5. Is the manuscript presented in an intelligible fashion and written in standard English?

Reviewer #1: Yes

Reviewer #2: Yes

6. Review Comments to the Author

Reviewer #1: I went over the revised version of the work. I am satisfied with the quantity and quality of adjustment that has been made. The manuscript looks considerably better and the flow of information is much easier to comprehend.

Reviewer #2: The author has made the necessary corrections based on my previous suggestion, and I agree with that. However, I would like to point out that there might have been a misunderstanding regarding my mention of a research framework. A research framework refers to a precise representation of the structure of a research project plan. By utilizing this "structure," one can determine the critical areas of the study. It would be highly beneficial to include a visual representation of the research framework, such as a diagram, in the paper. Doing so would significantly enhance the overall quality of the paper.

7. PLOS authors have the option to publish the peer review history of their article (what does this mean?). If published, this will include your full peer review and any attached files.

Reviewer #1: No

Reviewer #2: No

---

## [Editor Report · Acceptance letter]

3 Aug 2023

PONE-D-23-09529R1 

Influence of Psychological Capital on Core Competency for New Nurses 

Dear Dr. Xia:

I'm pleased to inform you that your manuscript has been deemed suitable for publication in PLOS ONE. Congratulations! Your manuscript is now with our production department. 

Kind regards, 

on behalf of

Dr. Chunyu Zhang 

Academic Editor

PLOS ONE